# AI-Powered Gamified Scaffolding: Transforming Learning in Virtual Learning Environment

Xuemei Jiang [1], Rui Wang [2,*], Thuong Hoang [1], Chathurika Ranaweera [1], Chengzu Dong [3,*] and Trina Myers [1]

[1] School of Information Technology, Deakin University, Burwood, VIC 3125, Australia; jiangx@deakin.edu.au (X.J.); thuong.hoang@deakin.edu.au (T.H.); chathu.ranaweera@deakin.edu.au (C.R.); trina.myers@deakin.edu.au (T.M.)

[2] CSIRO Data61, Eveleigh, NSW 2015, Australia

[3] Dvision of AI, School of Data Science, Lingnan University, Hong Kong, China

[*] Correspondence: r.wang@csiro.au (R.W.); chengzudong@ln.edu.hk (C.D.)

## Abstract

Gamification has the potential to significantly enhance student engagement and motivation in educational contexts. However, there is a lack of empirical research that compares different guiding strategies between AI-driven gamified and non-gamified modes in virtual learning environments to scaffold language learning. This paper presents an empirical study that examines the impact of AI-driven gamification and learning strategies on the learning experience and outcomes in virtual environments for English-language learners. A gamified English learning prototype was designed and developed. A between-group experiment was established to compare different gamified scaffolding groups: a traditional linear group (storytelling), an AI-driven gamified linear group (task-based learning), and a gamified exploration group (self-regulated learning). One hundred students learning English as a second language participated in this study, and their learning conditions were evaluated across three dimensions: engagement, performance, and experience. The results suggest that traditional learning methods may not be as effective as the other two approaches; there may be other factors beyond in-game interaction and engagement time that influence learning and engagement. Moreover, the results show that different gamified learning modes are not the key factor affecting language learning. The research presents guidelines that can be applied when gamification and AI are utilised in virtual learning environments.

**Keywords:** gamification; AI; transforming learning; virtual reality; education

## 1. Introduction

With the growing demand for foreign language learning and its positive impact on cognitive abilities, second-language education has been implemented into formal school curricula in many countries [1,2]. In first-language (L1) learning, the process occurs naturally through constant exposure to the language environment, with little to no formal instruction [3]. However, compared to L1 learning, second-language (L2) learning heavily depends on structured educational experiences in formal settings. Motivation emerges as a crucial factor for achieving L2 learning objectives, driving sustained effort and ensuring success across varying levels of difficulty [4,5]. Artificial intelligence (AI) has the potential to revolutionise L2 education by providing personalised, adaptive learning experiences that address individual challenges and maintain learner motivation. Integrating AI tools into

repetitive practice and memorisation tasks can help overcome one of the key challenges of L2 learning: staying motivated throughout the journey [6].

Despite students having intrinsic motivations for every challenge, sustained student efforts are required to gain positive outcomes, especially in language learning [7]. Prolonged effort without observable progress can drain learners' intrinsic motivation [6] and reduce their willingness to continue trying [7]. Learners lacking motivation could have less desire to learn, making it harder for them to achieve efficient learning [4]. Thus, discovering the right motivation is essential for language learners to overcome repetitive work along their learning journey and stay motivated to achieve their goals [4].

There has been a reasonable amount of work on the use of gamification to scaffold language learning in a blended environment. A review study by [8] on mobile-based language learning identified that the number of studies has grown from 0 records in 2000 to 699 records in 2016. However, the majority of the current works focus on why and how to apply game elements to scaffold language learning and the impact of digital game-based language learning (DGBLL) in language learning. A systematic review from [9] highlighted that from 2012 to 2017, research on the impact of digital games on L2 skills has been lacking, and more investigation is needed. Moreover, the research focus should shift from comparing only gaming and non-gaming conditions to other areas, such as the efficiency of DGBLL in L2 learning. Only a few studies have integrated game elements with a single learning strategy, and even fewer studies have compared two or more gamified learning strategies. A theoretical analysis and bibliometrics on game-based self-regulated language learning showed that the trend in GBLL (game-based language learning), examined within the broader context of SRLL (self-regulated language learning), began to emerge as a research trend between 2015 and 2020, with only 54 papers on GBLL of 314 papers on SRLL [10]. Future investigations are required on the underlying mechanisms, contingencies, and systematic design of technology-enhanced scaffolds and their corresponding effectiveness for supporting problem solving [11]. It is also identified that there has not been sufficient research that provides scientific evidence of the impact of different learning strategies with game elements as language-learning scaffolding.

The research work presented in this paper aims to fill this gap and investigates how different gamified scaffolding strategies may impact learning a language in a virtual learning environment through a scientific approach. To discover a suitable gamified scaffolding method for improving the student learning experience and outcomes in the virtual learning environment, the paper investigates how a gamified non-linear learning method (exploration mode) affects the learning experience and outcomes compared with both AI-driven gamified linear (linear mode) and traditional non-gamified and non-linear learning methods (traditional mode).

The main contribution of this paper is the examination of the correlation between the application of widely used game elements and L2 learning through a large-scale experiment that compares three levels of gamified strategies. Additionally, this paper provides guidelines for constructing future virtual language-learning environments that effectively utilise engagement, enhance learning experiences, and optimise learning outcomes.

- The research examines how various gamification modes affect learner engagement, experience, and perceived outcomes in a second-language virtual environment.
- Our study engaged a cohort of one hundred ESL (English as a second language) students, providing valuable insights by comparing different gamified levels of learning environments.
- The results of the study indicate that it is essential to understand user backgrounds to ensure effective virtual learning.

- While higher interactivity in virtual learning environments can enhance the learning experience, it might have a negative impact on actual learning outcomes.

  As a result, three hypotheses are proposed:

- H1: In the blended language-learning environment, learners are more engaged in learning with a non-linear gamified guiding mode (exploration mode) than in the other two learning modes.
- H2: In the blended language-learning environment, learners have a better learning experience in learning with a non-linear gamified guiding mode (exploration mode) than in the other two learning modes.
- H3: In the blended language-learning environment, learners perceive a better understanding of learning with a non-linear gamified guiding mode (exploration mode) than in the other two learning modes.

The rest of the paper is organised as follows. Section 2 provides an overview of the related work, covering learning strategies, L2 learning in virtual environments, and gamification. Section 3 presents the experiment design, flow, system structure, system development, and the methodology for data collection and analysis. Section 4 analyses the results, evaluates the proposed hypotheses, and discusses the factors influencing L2 learning. Section 5 discusses the findings compared with relevant past works, exploring aspects such as interactivity, motivation, and engagement. It also reflects on the limitations of this study and outlines potential future research directions based on the discussions and results. Finally, in the concluding Section 6, the results are summarised, and guidelines for applying gamified elements in the virtual learning environment (VLE) are suggested.

## 2. Related Work

The work presented in this paper comes from three research perspectives: learning strategies, learning language in the virtual learning environments, gamification and artificial intelligence. This section discusses and reviews the recent research presented in those areas.

### 2.1. Guiding Strategies in Learning

Scaffolding, or instructional scaffolding, refers to a teaching approach in which educators provide structured support during the early stages of learning and gradually reduce that support as students gain independence. It has been widely recognised as a critical component in facilitating student learning [12]. The scaffolding theory professes that such interactions in learning can significantly improve learning efficiency for students in learning a new concept. In traditional scaffolding, teachers support and teach in person, guiding students to focus on constructing knowledge to decrease the learning complexity. Along with technology development, the diversity of scaffolding concepts extends beyond the physical classroom [11].

Ref. [13] defined task-based learning (TBL) as an effective and efficient education strategy. Task-based learning ensures that the student achieves the learning objective and, at the same time, creates a learning environment that provides rich experiences. Therefore, the real experiences from TBL make it one of the major approaches to training students in the medical education field. A related approach, problem-based learning (PBL), also emphasises real-world engagement by encouraging students to focus on solving real-life problems and developing solutions, has also been proposed as a complementary strategy in this context [13]. In comparison to PBL, TBL focuses on the processes and the steps toward the result.

Different to TBL and PBL, self-regulated learning (SRL) heavily depends on learners exploring learning and being self-motivated to achieve the final goal. SRL is vital for successful learning in computer-based learning environments, and it is also a critical skill for people in lifelong learning [14]. In some research, SRL is called the exploratory learning model (ELM) or the discovery learning model. Regardless of the target demographic, usage context, choice of technology, and underlying pedagogy are highly related to the efficacy of any assessment approach. The result of applying the assessment approach differs when transferred to other groups of learners, and different contexts and educational situations [15]. Since the concept of using game design has been proven to affect the non-game context positively, scholars have demonstrated their interest in investigating the interdisciplinary field between game-based learning and SRL [10]. Ref. [10] considered the positive correlations between both strategies (SRL and DGBL) on students' motivation and self-efficacy [10]. The study showed that the student acquired enjoyment and a sense of control in game-based learning, driving the student's motivation and self-efficacy, leading them to sustained effort in learning to achieve higher levels of autonomy in self-regulated language learning (SRLL) [10].

### 2.2. Second-Language Learning in a Virtual Learning Environment

Along with technological innovation, the advantages of various digital settings in language learning have received escalating attention. To enrich the learning experience, researchers are investigating digital approaches that bring real-life situations into learning; such environments help learners deal with different life situations by coordinating theory and practice to consolidate understanding [16]. Past studies have created a learning environment that can offer students knowledge by experiencing it themselves, not just simply by trying to learn from others' experiences [16].

To fulfil the needs of diverse language learners [1], the learning environment should include more engagement and immersive design to maintain their motivation. Second-language acquisition researchers have also suggested in various studies that a learning environment that provides extensive reading can help to facilitate vocabulary learning and the development of various language skills [17]. A learning method that provided similar knowledge input to learners as extensive reading was also found in interactive adventure games, as they usually include a large number of dialogues and instructions performed by game characters in the target language, creating an authentic learning scenario for L2 learners [17]. Under this learning environment, learning is provided in an implicit way, and learners can acquire knowledge without deliberately learning.

Moreover, academics have investigated integrating different types of game environments to improve language learning. An analysis study stated that massively multiplayer games were the most commonly used genre in L2 learning [18]. Studies with positive feedback in L2 learning are mostly conducted in the Sims games and interactive story games [18]. Ref. [19] conducted a study that demonstrated the positive impact of integrating interactive music games with language learning on the acquisition of L2 vocabulary. Adventure games have also been suggested by researchers and many language learners, as they can be useful as a tool to increase learners' motivation and help them improve their language listening and speaking skills [3,17,20].

### 2.3. Gamification in Language Learning

Past studies mentioned that a gamified teaching–learning environment is an unusual, new, previously unknown thing that is created to arouse the interest and curiosity of the learners, thus stimulating the acquisition of new knowledge [16,21]. Gamification not only uses game elements in non-game environments but also attracts learners to empower their

engagement and motivation in the learning approach in a relaxed atmosphere [21]. The application of gamification theories in various fields is growing as an emerging field. A few investigations have also found that it is vital to develop a system with challenges and rules, interactivity, and feedback for the player to attempt to accomplish a goal that can bring out their emotional reactions [9,21]. The potential to shape users' behaviour in a desirable direction has driven gamification to be adopted rapidly in business, marketing corporate management, wellness, and ecology initiatives in recent years [22].

Many researchers have suggested that using gamification in the education environment provides an immersive game-like experience that facilitates game-like thinking and strategies for learners [23]. Moreover, due to the ability to teach and reinforce knowledge and skills, the use of educational games as learning tools is a promising approach [22]. In applying gamification in learning, the learner finds more reasons to engage and improve. Ref. [22] identified some of the most important elements behind gamification. For example, goals, challenges, customisation, progress, and feedback form the foundation for the learner's intervention plan, as they clearly fit the environment provided. Additionally, game elements are being applied to other parts of life, such as work and learning, with the presence of challenges, feedback, and reward contributing to an increase in enjoyment [24]. A recent analysis showed that a comprehensive user profile design is essential in future gamification learning. The gamification design of motivational affordance should cover a variety of user characteristics. For example, the achiever type of learner is influenced by their initial intrinsic motivation; they also tend to try new challenges with few or no physical rewards. While the socialiser is negatively influenced by learning content and motivated by social connections and relatedness [25].

Ref. [26] elaborates on differences of characteristics in order to design gamified learning. They use the Hexad framework [27] to classify gamification user types (achiever, disruptor, free-spirit, philanthropist, player, and socialiser). The Hexad framework is noted as a user classification specifically designed for gamification and highly suitable for personalising gamification experiences. The study examines perceived engagement on the ecological gamification design. The study introduces ecological gamification [28] as a novel design strategy related to environmental properties, and identifies its elements as chance, imposed choice, economy, rarity, and time pressure. The User Engagement Scale [29] was used to identify learners' engagement. The scale consists of 31 items and is purported to measure four dimensions of engagement: focused attention, perceived usability, aesthetic appeal, and reward. The results highlighted the influence of different characteristics on the usability of gamification elements and the importance of aligning gamification design with user type to maximise its effectiveness.

Over the past decades, scholars have revealed the positive impact of digital games on improving vocabulary, grammar, writing, and speaking in language learning [20]. A recent review highlighted that 59 studies from 2000 to 2018 indicated that vocabulary was the most frequently examined skill in language learning at around 50% [20]. This was followed by overall language proficiency at around 15%. Another recent review from [9] explored 49 DGBLL studies from 2012 to 2017. In the study, one of the results revealed students' positive attitudes toward using DGBLL in language learning. Around 63% of studies investigated student perspectives of the DGBLL experience, and about half of the studies found that learners expressed a positive attitude toward the learning game. DGBLL supports learners in effectively overcoming the learning barrier, reduces their language-learning anxiety [20,30], and increases their willingness to communicate [18].

Despite the possibilities and benefits of gamification in the educational context, there are a few common concerns and gaps. The study in [19] uses interactive music games to hinder the acquisition of second-language vocabulary. In this study, the authors investigated eighty undergraduates paired based on language skills and game proficiencies [17]. One subject played an interactive music game for vocabulary learning; the paired subject watched the game simultaneously on another monitor. The result showed that the player group recalled less vocabulary from the game than the watcher group. This finding pointed out the difficulty of simultaneously paying attention to gameplay and vocabulary learning. A recent study in [31] discovered that this finding matched with statements from previous studies. Ref. [31] developed a VR-based second-language-learning tool with three different levels of interactivity (no interactivity, low interactivity, and high interactivity). The study invited 56 non-Spanish speakers to three interactivity-level groups by taking a questionnaire and pre- and post-study evaluations to compare their learning outcomes, learning experiences, and other factors affecting learning. The results showed that higher interactivity has a negative impact on learning outcomes. On the other hand, higher interactivity positively affects the learning experience. This result from [31] aligns with the observation from [32]. In addition, ref. [32] identified that users driven by external motivations, like achieving a specific goal with their activity tracker, often opt for less effort when the tracker is not present, leading to a heightened reliance on the device. However, those who engage in physical activity for enjoyment or intrinsic reasons face a lesser decline in motivation in the tracker's absence.

Further, several research works identified some challenges in using gamification in the educational context. Ref. [22] pointed out that proper empirical research on the effectiveness of incorporating game elements in the learning environment is still scarce. Most of the studies describe game mechanics and dynamics and reiterate their possibilities in educational use. The lack of practical use causes unbalanced integration of the game elements, learning content, and learning environment. Existing studies mainly focus on classic gamification designs like points, badges, and leaderboards, which might be the reason for the mixed results in the broader gamification literature [26]. Past studies stated that the integration of the games, the educational method, the game design, and the content needs to be solved to be able to contribute to effective learning [16]. By creating a learning environment that mimics the interactive and immersive aspects of games, educators can stimulate students' intrinsic motivation, encouraging them to engage more deeply with the material [10,16,33]. The effectiveness of gamification in education is nuanced, with studies indicating that its impact may vary based on the students' intrinsic motivation levels and the specific design of the gamified system [25,34]. Additionally, the systematic review presented by [9] highlighted that future investigation is necessary to achieve efficient L2 learning in a digital environment setting. The focus should shift from comparing gaming and non-gaming conditions to the efficiency of DGBLL in L2 learning.

### 2.4. Artificial Intelligence in Education

Past research emphasises the transformative potential of AI and big data in the educational field, proposing that these technologies are key to developing learner-centred, personalised approaches that fulfil individual student needs. AI and big data enable the meticulous collection and analysis of student data, facilitating timely and accurate assessments of learner performance [35]. These intelligent educational systems accelerate the personalised process to identify unique learning patterns and needs, ensuring that education is tailored to each student [36]. AI is increasingly recognised as a critical force in the "Fourth Industrial Revolution" and is becoming an essential component of educational curricula [36,37], reflecting its growing influence in both e-learning and traditional

settings. The differentiation between online and offline educational methods highlights the challenges in evaluating student learning, which AI and big data can address by adapting techniques from other fields [37]. Over time, AI has evolved to include various technologies like machine learning, natural language processing, and neural networks [38]. Initially, AI in education focused on developing intelligent tutoring systems to enhance educational outcomes and operational efficiency [36]. The ongoing developments in digital resources, gamification, and personalised learning experiences provide numerous opportunities for expanding the application of AI in education, suggesting a dynamic future for this field. Although the purported benefits of AI in education are great, many concerns have been raised. Ref. [36] highlighted that the current AIED system mostly relies on learning patterns based on the averages of prior learners, which limits its effectiveness for being truly tailored to individuals.

### 2.5. AI-Powered Gamification in Education

The convergence of AI-driven personalisation and gamification presents a promising avenue for addressing the challenges of student engagement in the digital age. By leveraging AI to tailor gamified learning experiences to the individual needs and preferences of students, educators can create highly engaging and effective educational environments. This approach not only capitalises on the motivational benefits of gamification but also ensures that the learning experience is aligned with the cognitive and emotional needs of each student, thereby maximising the potential for deep and meaningful learning [39,40].

The transition to digital learning environments has highlighted the need for more engaging, effective, and personalised educational experiences. The introduction of AI-driven personalised gamified environments in educational contexts could significantly enhance the learning experience by providing more engaging, adaptive, and efficient educational tools. The intersection of AI and gamification in education has undergone significant transformation, particularly following the public release of the first ChatGPT version used GPT-3.5 model in November 2022 by OpenAI [41]. Early studies focused on intelligent tutoring systems, but current work emphasises AI as a collaborative tool that empowers teachers and personalises student learning pathways [36]. This technology is expected to analyse a student's learning habits, strengths, and weaknesses in real time, offering highly customised content that keeps students challenged but not overwhelmed. Integrating AI with gamification elements can further motivate students by making learning fun and rewarding. Such environments can also facilitate immediate feedback, enabling students to understand their mistakes and learn from them promptly [42]. Ref. [43] exemplifies this trend with their development of GAMAI, a tool that generates gamified programming exercises through OpenAI's API. Their tool integrates with the Framework for Gamified Programming Education (FGPE) and has been positively received by students for its usability and engagement. Ref. [44] extended this paradigm by integrating AI into immersive learning scenarios in VR and AR. Their case studies highlight real-time adaptation of difficulty, personalised learning paths, and dynamic feedback in virtual STEM simulations. Moreover, ref. [42] conducted a large-scale SEM analysis with 290 university students to evaluate the impact of AI gamification on engagement and achievement. The results confirmed statistically significant relationships between AI, gamification, and academic performance.

AI-driven solutions can offer even more personalised approaches, adjusting the learning material's complexity and presentation style to suit each student's unique requirements. Overall, integrating AI-driven personalised gamified environments into educational contexts could lead to improved student engagement, faster learning progress, and higher overall satisfaction with the educational experience.

## 3. User Study

To fill the research gaps, we designed an experiment aimed at addressing these limitations. Our study involved 135 participants, who were assigned to three distinct levels of a gamified virtual language environment. We evaluate the proposed hypotheses by collecting and analysing their in-game activities and designing a post-experiment survey to assess their experience, and examining the factors that influence their engagement, learning experience, and perceived learning outcomes.

A proof-of-concept (PoC) prototype application (named SimpleLingo) has been designed and developed to facilitate learners to learn English as an L2 in mobile-based 3D interactive environments. The scenario-based learning method is adopted, and a café scenario is used as a case study. Dialogues and activities are designed to cover various topics in a café environment, with an embedded avatar system enabling users to interact with the virtual environment. The learning scenario is a conversation in a cafe. The prototype includes 19 quests. Learners can learn English words related to the cafe, including coffee types, serving size, method of payment, and the Australian currency. The learning content includes introducing how to play the game and learning how to order in the cafe, and how to pay and chat with a friend in the cafeteria. Once learners start the learning session, the in-app instructions guide them on how to control, continue, and finish the session.

The prototype provides role-play functions for the participants to embody in-game characters, engage in conversations, and interact with non-player characters (NPCs) using speech. For a better immersive experience and to practise pronunciation, the role-play section uses Watson speech recognition to detect the user's voice and try to match it with the original text. Learners can choose to act as the barista or the customer to practise the pronunciation of the sentences. Two augmented reality (AR) mini-games are embedded during the learning session to help learners reinforce their learning with the advanced technology. In this prototype, the AR games are designed to enhance the memorisation of the learning materials through practice.

### 3.1. Participant

The participants of this study were 135 international students whose first language is Chinese and who are learning English as a second language. The invitations were distributed through university social media groups. The participants were recruited by reacting to the invitation through social media groups and research recruiting platforms. The participants' ages ranged between 18 and 30. They were randomly divided into three groups: Traditional, linear, and exploration, and each group included around thirty participants. The experiment lasted a month, from May 2022 to June 2022; participants could join the experiment by downloading the application on their own smartphone devices, and attend in their free time within the period.

### 3.2. System Design

The prototype system SimpleLingo (see Figure 1) was developed using the Unity game engine using the Game Creator 2 package. The Combu 2 package was used as a full-featured framework for the backend for managing and transferring the data to the server. During the role play, the Watson speech recognition API was used for the speaking practice game and to evaluate the user's performance. For assessing the user's speaking, the user's speaking input is converted to text by using Azure Speech to Text, then compared with the original text. Two AR mini-games were also added as part of this prototype to enhance the game-like experience. For the AR game, the EasyAR Sense 4.5.0 package is used for development.

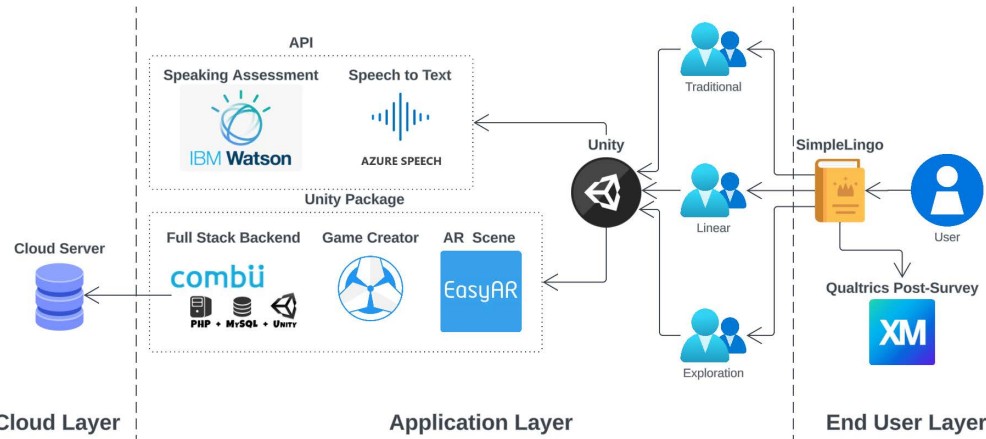

**Figure 1.** System architecture.

### 3.3. Procedure

This research has a between-group study design since the learning content for each group is the same, and every participant only attends the experiment once. This application gives the participant a unique participant ID. Our prototype randomly and equally divided the participants into three groups by the participant ID: Traditional, linear and exploration (see Table 1). Participants in each group then needed to conduct the learning session using the application. The learning session for each group included the tutorial, 19 quests, speech recognition role-play practice, and two AR mini-games.

**Table 1.** Experiment group design.

|  | Game Mode | | |
| --- | --- | --- | --- |
|  | Traditional | Linear | Exploration |
| Full Learning Content | ✓ | ✓ | ✓ |
| Role-play Practice | ✓ | ✓ | ✓ |
| Movement Control | ✗ | ✓ | ✓ |
| Interactive Environment | ✗ | ✓ | ✓ |
| Progress Control | ✗ | ✗ | ✓ |

#### 3.3.1. Traditional Group

For the traditional group, the learning content is presented in a linear way with minimal interactions, as shown in Figure 2, a screenshot taken from the application. The traditional group has a traditional online learning environment in the game that only includes minimal interaction. Learners are not allowed to change or skip chapters and can only follow the quests step by step. Learners cannot control the movement of the in-game character, and the character automatically moves to the quest position or quest item to start the next quest. Learners in this traditional group only need to touch the conversation window to continue the conversation and practise their speaking in the speech recognition role-play game. The learner in this group is not permitted to change the learning chapter or check all the chapters and their current progress. Therefore, instead of a chapter panel, a progression bar is shown at the top of the screen to help them understand the current progression.

#### 3.3.2. Linear Group

For the linear group, the learning content is presented in a linear way, guided by a virtual AI avatar with many interactions. A screenshot of the application used for the linear group is shown in Figure 3. Compared to the traditional game mode, in the linear mode, the content is guided by an AI avatar. Participants are allowed to control the

character's movement through a joystick controller, explore the environment, and interact with environmental items using touch controls according to the quest requirements. The screenshot in Figure 3 shows that the interactable quest item is outlined in the learning environment; learners can explore the environment to find and touch the item to continue the quest. Like the traditional group, the learning in this game mode is linear, thus learners are not allowed to control the quest progress, or skip or change the chapter. The quest progress is displayed on the screen to inform them of the progress towards the goal.

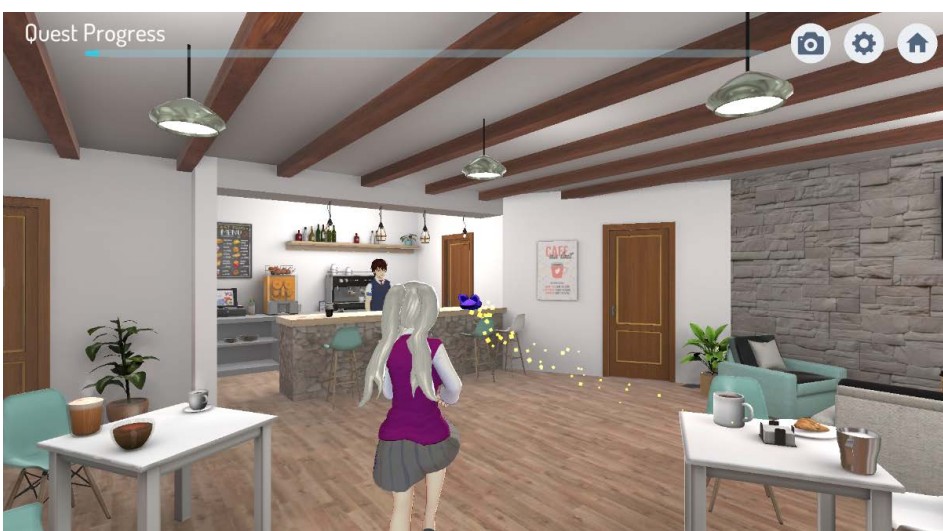

**Figure 2.** Traditional game mode.

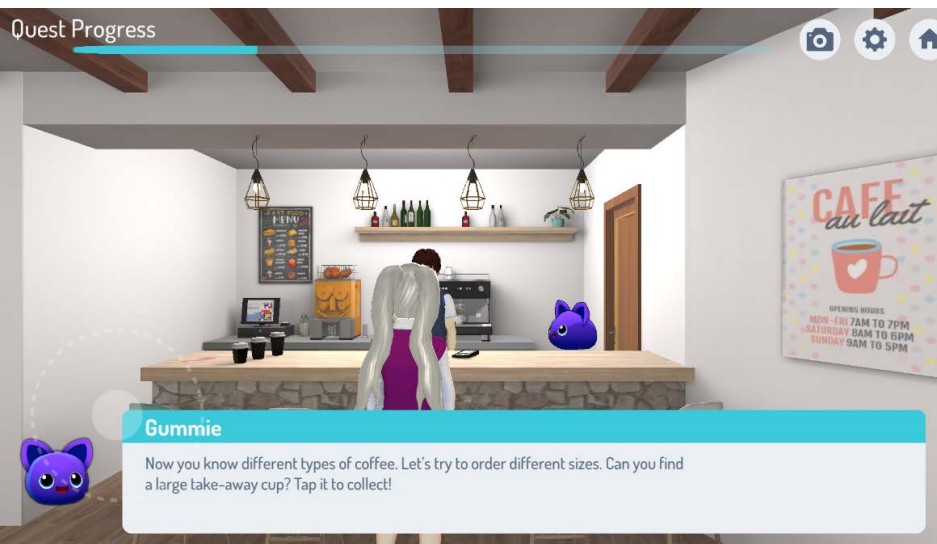

**Figure 3.** Linear group.

### 3.3.3. Exploration Group

The exploration group offers a non-linear learning experience with multiple interactions, as shown in Figure 4. Participants can navigate the character using a joystick controller, granting them autonomy over their learning journey. Compared to the linear game mode, this approach empowers learners with the freedom to control their quest progress. There is a "current quest" button on the top left of the screen. Through this button, they are able to access the quest panel, which allows them to change, skip, or restart any chapters.

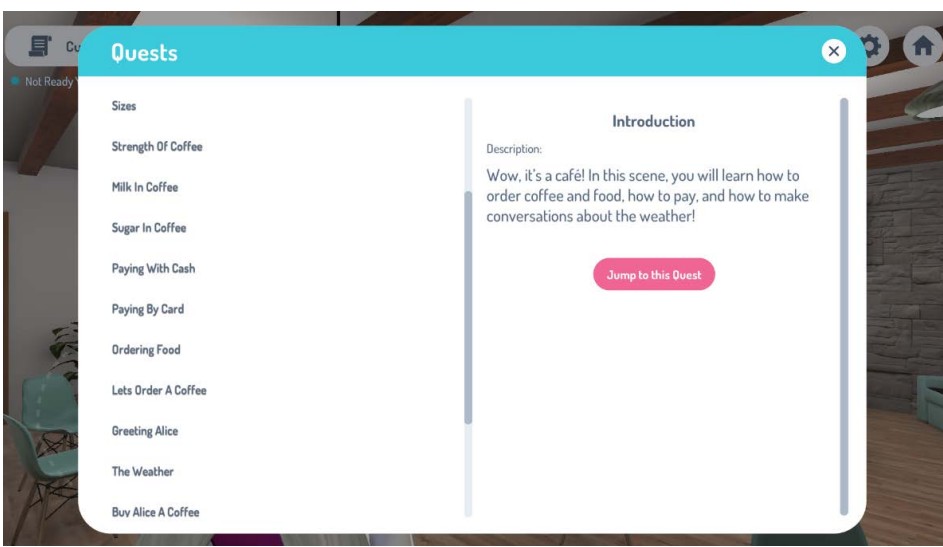

**Figure 4.** Exploration group.

3.3.4. Data Collection and Analysis

For data collection and analysis, this study employed two methods: a post-experiment survey and in-game data collection. Gender was excluded from the data, as previous research, such as [31], suggested it is not a pivotal factor in VLE for L2 studies. The post-experiment survey includes 4 general questions, 18 Likert-scale questions, and an open-ended question (see Table 2). The participants attempt the questions in sections, as shown in Table 2. The post-experiment survey data mainly use a 5-point Likert scale to indicate how much they agree with each statement (strongly [dis]agree, somewhat [dis]agree, and neutral). The post-experiment survey items were selected from various questionnaires to determine factors affecting language learning. Questions 1, 2, 3, and 4, sourced from the public speaking anxiety scale (PSCAS) questionnaire [45], assess the impact of foreign language anxiety on learning proficiency. Questions 7, 9, 11, 12, and 14, adapted from the IEQ questionnaire [46], focus on users' perceptions of learning progress, involvement, immersion, and guidance, aligning with the study's purpose. The satisfaction and self-confidence in learning (SCL) questionnaire [47] provided questions 8, 13, 15, and 17, gauging the perceived value of the training for language learning. Questions 6, 10, and 16, taken from the training evaluation questionnaire [48], assess satisfaction with the training, engagement, and ease of following the learning. Lastly, questions 5 and 18, adapted from the Assessment Usability and Educational Practices Questionnaire (EPQ) [49], measure perceived difficulty and willingness to recommend the learning activity. In this research, for collecting in-game data, C# scripts were written to manage the data transfer, collection, and recording. The in-game data collection method captures user interactions throughout the game. This encompasses metrics such as "game mode", "current quest", "skip count", and "role-play count". Specifically, the "skip count" denotes how often a user opts to bypass a section, and the "role-play count" indicates the frequency of users practising conversations in a role-playing scenario. Data on the "game mode" and "current quest" provide insights into the game segments users are actively engaging with. The time-spent section collects the data related to time, such as "time spent" and "timestamp".

The post-experiment survey data underwent simplification, resulting in five distinct factors, using the KMO and Bartlett's test. With a KMO measure of sampling adequacy at 0.835 and a *p*-value below the set alpha level of 0.05, it was determined that factor analysis was appropriate. The five factors were learning experience level, English anxiety level, learning difficulty level, and learning guidance level. In Table 3, the questions highly correlated to the factor are marked in grey. According to the factor analysis results,

questions 9, 10, 11, 12, 13, 14, 17, and 18 were combined into factor 1, named the "Learning Experience Level" since all the questions included in this factor are related to the learning experience. Questions 1, 2, and 3 were combined into factor 2. These three questions are related to English anxiety; therefore, factor 2 was named the "English Anxiety Level". Questions 4 and 5 in factor 3 are related to the difficulty of the learning; therefore, they were defined as the "Learning Difficulty Level". Factor 4 included questions 6, 7, and 8, and was named the "Learning Confidence Level". Questions 15 and 16 were combined into factor 5, named the "Learning Guidance Level".

**Table 2.** Post-experiment survey.

| Questions |
| --- |
| General questions |
| What is your ID on the home screen? |
| What is your first language? |
| Which country are you living in now? |
| Please indicate your self-assessed English level |
| Survey questions |
| (1) You get nervous and confused when you are speaking English. |
| (2) You want to speak less because you feel shy while speaking English. |
| (3) You start to panic when you have to speak English without preparation in advance. |
| (4) You feel less anxious to speak English after you have practised and prepared the conversation. |
| (5) Rate the difficulty you experienced in carrying out the task |
| (6) The contents/materials distributed were helpful. |
| (7) You were making progress toward the end of the session. |
| (8) You are confident that you are mastering the content of the simulation activity that the training presented to me. |
| (9) Overall, how interested are you in the learning session? |
| (10) Participation and interaction were encouraged in the learning session. |
| (11) You were motivated to continue learning new content in the learning session |
| (12) Did you find yourself becoming so involved that you were unaware you were even using controls at any point in the learning session? |
| (13) The teaching methods used in this training were helpful and effective. |
| (14) Overall, you were guided toward the end of the session. |
| (15) You know how to get help when you do not understand the concepts covered in this training. |
| (16) Overall, the learning session was easy to follow. |
| (17) Did you enjoy the learning session? |
| (18) You would recommend the learning materials to your friends or classmates. |
| Do you have any other comments regarding the questions above? |

**Table 3.** Post-experiment survey factor analysis structure matrix result [1,2].

| | Component | | | | |
| --- | --- | --- | --- | --- | --- |
| | **1** | **2** | **3** | **4** | **5** |
| 1 | −0.028 | 0.897 | −0.041 | 0.009 | 0.054 |
| 2 | −0.063 | 0.926 | −0.071 | 0.000 | −0.117 |
| 3 | 0.177 | 0.813 | −0.041 | −0.159 | −0.011 |
| 4 | 0.293 | 0.117 | 0.752 | −0.047 | −0.236 |

**Table 3.** *Cont.*

|  | Component | | | | |
|---|---|---|---|---|---|
|  | **1** | **2** | **3** | **4** | **5** |
| 5 | 0.111 | 0.209 | −0.789 | 0.068 | −0.254 |
| 6 | 0.554 | 0.125 | −0.023 | −0.835 | −0.074 |
| 7 | 0.547 | 0.072 | 0.016 | −0.891 | 0.093 |
| 8 | 0.274 | 0.044 | 0.307 | −0.704 | 0.557 |
| 9 | 0.844 | 0.174 | 0.106 | −0.501 | 0.128 |
| 10 | 0.813 | 0.179 | 0.134 | −0.579 | 0.091 |
| 11 | 0.873 | 0.148 | 0.177 | −0.543 | 0.111 |
| 12 | 0.792 | −0.058 | 0.092 | −0.362 | 0.031 |
| 13 | 0.890 | −0.012 | 0.072 | −0.337 | 0.132 |
| 14 | 0.777 | −0.005 | 0.094 | −0.480 | 0.217 |
| 15 | 0.556 | 0.117 | −0.073 | −0.185 | 0.721 |
| 16 | 0.571 | −0.081 | 0.358 | −0.433 | 0.608 |
| 17 | 0.832 | 0.008 | 0.031 | −0.417 | 0.309 |
| 18 | 0.884 | 0.075 | 0.046 | −0.333 | 0.216 |

[1] Extraction method: Principal component analysis. [2] Rotation method: Oblimin with Kaiser normalisation.

# 4. Results

This study used a post-experiment survey and in-game data to assess the hypotheses that the three components, engagement level (H1), learning experience (H2), and perceived learning (H3; determined by the factor "Learning Confidence Level"), had a significant effect based on the learning groups. The study had specific criteria for valid responses: matching participant IDs in-game and in the survey, completion of the learning session, and a minimum learning time of 1200 s. These ensured participants fully experienced the session before survey completion. Despite precautions, some mismatches occurred due to forgotten login information or multiple device usage. After filtering all the mismatched responses, 100 valid responses remained from 125 received responses (T = 36, L = 34, E = 30). Additionally, to investigate the first hypothesis in detail, the engagement level was divided into five factors. H1.1: student-perceived engagement level (determined by the factors "Learning Difficulty Level" and "Learning Guidance Level" in the following content), based on Flow theory; H1.2: student-performed interactions; H1.3: student-performed positive interactions; H1.4: student-performed negative interactions; and H1.5: time spent on learning. H1.2, H1.3, and H1.4 pertain to student-performed interactions, both positive and negative, counted in-game. H1.5 refers to time spent on learning, indicating attention to learning. In this experiment, all the null hypotheses were accepted at a 95% confidence level, and all the quantitative results revealed no statistical differences between the groups' learning engagement, experience, and performance. The quantitative results are discussed in the following section.

## 4.1. The Impact of Gamified Scaffolding on L2 Learning

The five factors were analysed using one-way ANOVA, and the results (as shown in Table 4) show that none of them rejected the null hypothesis, which means there was no significant difference between these three groups from the engagement aspect. However, the interpretation based on the histogram, box plots, and mean plot graphs can provide valuable information. From the post-experiment survey box plots (see Figure 5), there are several outliers across the five factors, where the linear group has the most outliers and the traditional has the least. The outliers reveal that the learners in the linear group perceived the most diverse overall learning experience, followed by the exploration group, with the traditional group last.

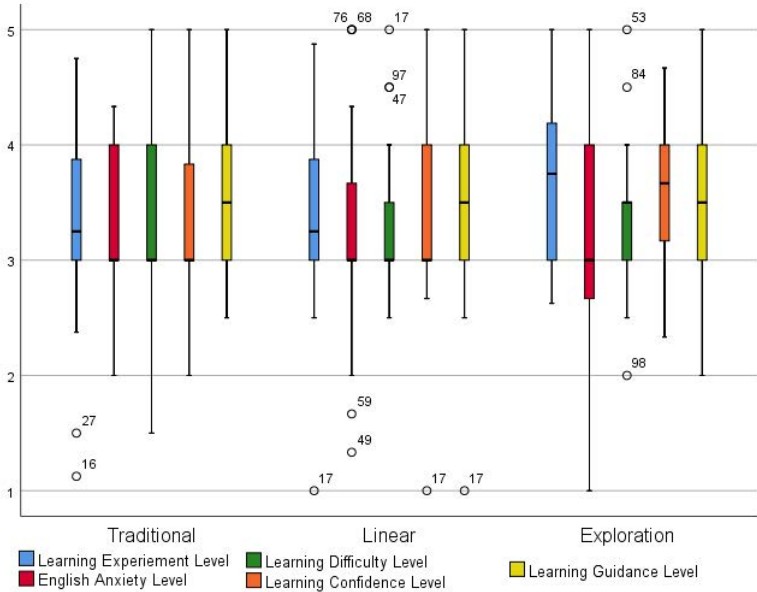

**Figure 5.** Box plots for post-experiment survey results. The numbers in the figure represent the Outliers.

**Table 4.** Summary of H1 results.

|  | H1.1 | H1.2 | H1.3 | H1.4 | H1.5 |
|---|---|---|---|---|---|
| Normality | $<\alpha = 0.05$ | $<\alpha = 0.05$ | $<\alpha = 0.05$ | $<\alpha = 0.05$ | $<\alpha = 0.05$ |
| K-W ANOVA | $>\alpha = 0.05$ | $>\alpha = 0.05$ | $>\alpha = 0.05$ | $>\alpha = 0.05$ | $>\alpha = 0.05$ |
| Descriptives | No Diff | No Diff | No Diff | No Diff | No Diff |
| Homogeneity | $>\alpha = 0.05$ | $<\alpha = 0.05$ | $>\alpha = 0.05$ | $>\alpha = 0.05$ | $>\alpha = 0.05$ |
| One-way ANOVA | $>\alpha = 0.05, F < 0.05$ | $>\alpha = 0.05, F > 0.05$ | $>\alpha = 0.05, F > 0.05$ | $>\alpha = 0.05, F > 0.05$ | $>\alpha = 0.05, F > 0.05$ |
| Turkey HSD | $<\alpha = 0.05$ | N/A | N/A | N/A | N/A |
| Power Calculation | $>\beta = 0.8$ | $>\beta = 0.8$ | $>\beta = 0.8$ | $>\beta = 0.8$ | $>\beta = 0.8$ |
| Null Hypothesis | Accepted | Accepted | Accepted | Accepted | Accepted |

N/A indicates that the data is not applicable or not available for this condition.

The results of the interactions (as shown in Figure 6 and Table 5) show that from the interactivity perspective, learners in the linear mode engage more in learning. However, this might be because of the two outliers in the linear mode's results, since the linear mode has almost double the standard error in the total and positive interactions as the other two modes. In comparing the time spent on learning, the data in all three modes are in a similar range, with the learning time distributed around 2000 s (as shown in Figure 7).

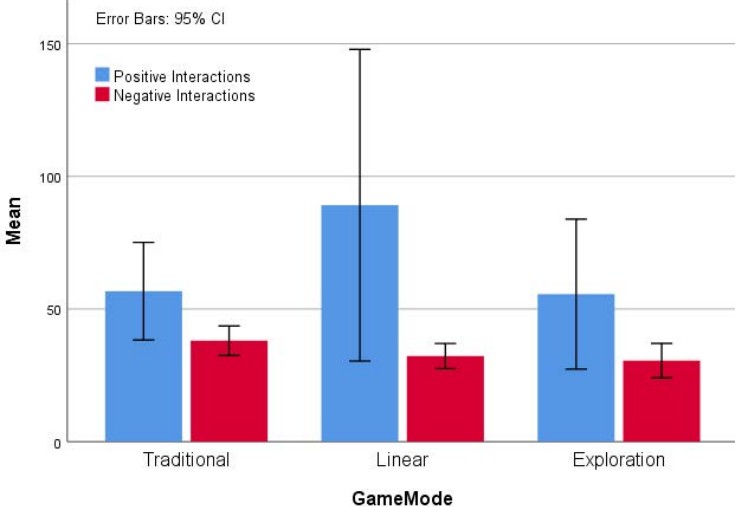

**Figure 6.** Comparison of group interactions.

**Table 5.** Summary of in-game interaction results.

|  |  | N | Mean | Std. Deviation |
|---|---|---|---|---|
| Positive Interactions | Traditional | 36 | 56.69 | 54.393 |
|  | Linear | 34 | 89.12 | 168.374 |
|  | Exploration | 30 | 55.60 | 75.664 |
|  | Total | 100 | 67.39 | 111.440 |
| Negative Interactions | Traditional | 36 | 38.03 | 16.455 |
|  | Linear | 34 | 32.26 | 13.583 |
|  | Exploration | 30 | 30.53 | 17.336 |
|  | Total | 100 | 33.82 | 15.994 |
| Total Interactions | Traditional | 36 | 94.72 | 56.996 |
|  | Linear | 34 | 121.38 | 171.438 |
|  | Exploration | 30 | 86.13 | 80.451 |
|  | Total | 100 | 101.21 | 114.304 |
| Time Spent (sec) | Traditional | 36 | 2139.89 | 1005.020 |
|  | Linear | 34 | 2057.65 | 914.918 |
|  | Exploration | 30 | 2019.73 | 803.709 |
|  | Total | 100 | 2075.88 | 909.887 |

As mentioned above, hypothesis H2 about the learning experience was rejected, using ANOVA to compare the data calculated from related questions. The visualisation of the test (as shown in Figure 5) indicates only a minor gap between the three groups (F = 1.923, $p$ = 0.152). The learners who participated in the experiment had a moderate attitude toward the learning experience in the prototype. According to the box plot results, all three modes have similar medians and a wide interquartile range (IQR). Outliers (user IDs 16, 27) in the traditional mode had a very negative attitude toward the experience. Despite AI guidance and more interaction, the perceived learning experience level in the linear mode was similar to the traditional mode. An outlier (user ID 17) in linear mode perceived the learning as of a low level of difficulty, with low confidence and low guidance. Outliers (user IDs 98, 84, 53) in exploration mode show a diverse perception towards the difficulty level. The presence of these outliers in the experiment indicates variability in individual experiences. The learning experience is the most challenging element to assess, including challenges such as prior learning experiences [50]. A survey about factors influencing EFL learning in China conducted by [51] pointed out that the learning experience reflects the learner's attitude toward L2 learning. Although this greatly depends on their past learning experience, it is affected by many other factors by learners themselves. For example, the degree of development of visual skills and different learning contexts [51]. Therefore, the theoretical nature of the L2 learning experience construct has not yet been specified [52]. Figure 5, comparing the learning confidence, shows that learners in exploration mode are more confident that they gain understanding throughout the learning than those in other modes. In contrast, learners in the traditional group have the lowest confidence about mastering the content they learned. However, Table 6 indicates that there is only a small gap between the three groups' means ($\mu$ T = 3.25, $\mu$ L = 3.37, $\mu$ E = 3.52). The ANOVA result for H3 was not significant at the 0.05 level, with an F-value of 2.027 and a $p$-value of 0.137. This indicates that there is not substantial evidence to suggest a difference in the perceived understanding across the three study groups in the learning domain.

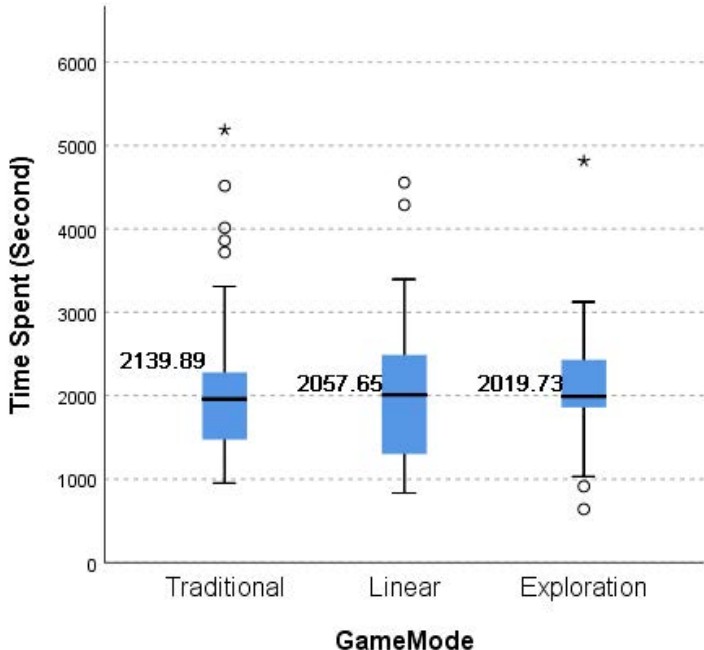

**Figure 7.** Comparison of time spent on learning in three groups. The asterisks (*) represent extreme outliers in the data.

Although the ANOVA tests did not highlight significant disparities between the groups, examining the nuances revealed some noteworthy insights. For instance, when considering the difficulty and guidance levels (as presented in Table 6), the traditional group reported the least challenges and felt they received the most guidance. According to gamification theory, there is a positive correlation between learning difficulty (or challenge) and guidance (encompassing instant feedback and formative assessment) with engagement levels. The linear group perceived an overall high difficulty with the lowest guidance. On the other hand, the mean of the exploration group showed the greatest difficulty with only slightly lower guidance than the traditional group. Therefore, the results showed that learners in the exploration group had a moderately higher perception of learning engagement compared to the other two groups. An intriguing observation from the comparison of total interactions, both positive and negative (as shown in Figure 6), is that learners in the linear mode are more inclined to engage with and enjoy the game's features and environment. The data indicates that the frequency of interaction is much higher than for the other modes, maybe due to the learner's inability to control the avatar meant for exploring and interacting with the environment in each quest. Nevertheless, learners in the traditional group are the least likely to enjoy the learning, as nearly one-third of their interactions were negative. Since they are not allowed to control both the movement and learning progress, significantly decreasing their interest in learning, they are more likely to skip the practice and finish the learning session as soon as possible.

The extreme numbers from the total, positive, and negative (Table 5) interactions may vary due to the technical issues in the prototype. However, to prevent the learners from skipping all the learning sections and going directly to the final chapter, the prototype in this study set a time requirement before the learner attempted the survey. Therefore, the time spent on learning for all groups is quite similar as the learner must reach the time required for the study. In the time spent on learning, shown in Figure 7, the traditional group achieved the highest mean in time spent on learning among the three groups; meanwhile, learners in the exploration group spent the least time learning. This result could be because learners in the exploration mode are allowed to skip the quests, as they can even directly

jump to the last chapter, while the traditional group cannot control the quest progress, they can only move through all the chapters step by step.

In the learning experience-level comparison, the exploration group reported the highest perceived learning experience. The traditional group was in second place, followed by the linear group having the lowest perceived learning experience. The exploration group having the highest perceived learning experience is an inevitable outcome, as learners can freely control and interact in this mode. Interestingly, despite the traditional group having fewer accessible features than the linear group, it still outperformed the linear group in perceived learning experience. The result of comparing the confidence level showed an approximately straight line from the lowest, the traditional group, to the highest, the exploration group. Learners in the exploration group reported being more confident in mastering the learning compared to the other two groups, followed by the linear group, and the last group was the traditional group. There was a 0.16 gap between the exploration group and the traditional group. In the traditional group, learners had no control over their learning progress and movements. Hence the default learning pace might not apply to them.

In summary, compared to the other groups, learners in the traditional mode engage less, performing moderate total interactions but with a high rate of negative interactions. Learners in the exploration mode perform the least total interactions; however, they perceive the highest engagement and the least negative interactions while learning. The linear mode graphs demonstrate that these learners had total and positive interactions higher than the other groups, yet had a low perceived engagement level.

**Table 6.** Summary of post-experiment survey results.

|  |  | N | Mean | Std. Deviation |
|---|---|---|---|---|
| Experience | Traditional | 35 | 3.3714 | 0.78969 |
|  | Linear | 34 | 3.3750 | 0.74366 |
|  | Exploration | 31 | 3.6935 | 0.72875 |
|  | Total | 100 | 3.4725 | 0.76264 |
| English Anxiety | Traditional | 35 | 3.2571 | 0.70981 |
|  | Linear | 34 | 3.1961 | 0.92521 |
|  | Exploration | 31 | 3.2043 | 0.94154 |
|  | Total | 100 | 3.2200 | 0.85309 |
| Difficulty | Traditional | 35 | 3.3286 | 0.70651 |
|  | Linear | 34 | 3.3382 | 0.59950 |
|  | Exploration | 31 | 3.3387 | 0.62433 |
|  | Total | 100 | 3.3350 | 0.63982 |
| Confidence | Traditional | 35 | 3.3048 | 0.66848 |
|  | Linear | 34 | 3.4216 | 0.80949 |
|  | Exploration | 31 | 3.6452 | 0.57048 |
|  | Total | 100 | 3.4500 | 0.70013 |
| Guidance | Traditional | 35 | 3.5857 | 0.65849 |
|  | Linear | 34 | 3.4412 | 0.83271 |
|  | Exploration | 31 | 3.5806 | 0.77564 |
|  | Total | 100 | 3.5350 | 0.75296 |

### 4.2. The Impact of Second-Language Anxiety on L2 Learning

As mentioned in the previous section, the participant population is of Chinese background, and the factor of second-language anxiety among Asian background language learners can also affect individual language-learning outcomes and experiences [2,53]. The second-language anxiety factor is considered and discussed to obtain more accurate results for the proposed hypotheses.

The descriptive Table 6 does not show significant differences in language anxiety levels between the groups. In the variances and ANOVA tests, the second-language anxiety factor between the groups delivers a similar result. Especially in both the ANOVA tests, the *p*-values are 0.942 (Kruskal–Wallis, one-way ANOVA) and 0.950 (one-way ANOVA) at the 0.05 confidence level, which means the three groups only have less than 0.058 statistical differences at the 0.05 confidence level. Therefore, the effect of second-language anxiety on the provided hypotheses can be omitted for this research. However, an interesting finding in this study is that the group with the highest language anxiety level had the lowest confidence in learning (see Table 6). Although the group with the lowest language anxiety performed most interactions in the experiment (see Figure 6), their learning engagement and experience (see Tables 4 and 6) results were insignificant. Two participants who commented negatively about their English learning also rated themselves with high scores on language anxiety questions. This result is the same as in the study by [30]: that there is no correlation between performance and language anxiety.

### 4.3. The Impact of Other Factors on L2 Learning

The qualitative data collected by the open-ended question at the end of the post-experiment survey were analysed. In 100 valid responses, 33% of participants left comments and feedback, as the open-ended question in the post-experiment survey was not compulsory. The survey was written in both Chinese and English. Chinese comments were translated into English for analysis. Figure 8 shows that learners in the traditional group were more likely to express their thoughts, based on the total number of comments, compared to the other two groups.

Thematic keywords from the comments were extracted. There is 15% positive feedback within the total feedback provided by the participants. As shown in Figure 9, there are 12 comments about the control method. The control of the view and the movement in the prototype affect users' experience the most. Some participants noted problems with the sensitivity and smoothness of both the view perspective control and the movement joystick control. The control method was discussed the most within the traditional group; around 42% of the comments were from the traditional group in 12 statements. One of the participants from the traditional mode group mentioned the traditional linear control, which does not allow them to move around freely, greatly affecting their experience. The second highest element was the speech recognition function. The eight total comments about speech recognition include 62.5% of feedback from the traditional mode, 12.5% from the linear mode, and 25% from the exploration mode. Most of the comments were about the accuracy of the assessment and the sensitivity of voice detection and recognition. A participant from the traditional group mentioned that the speaking accent makes it hard to correctly recognise the words.

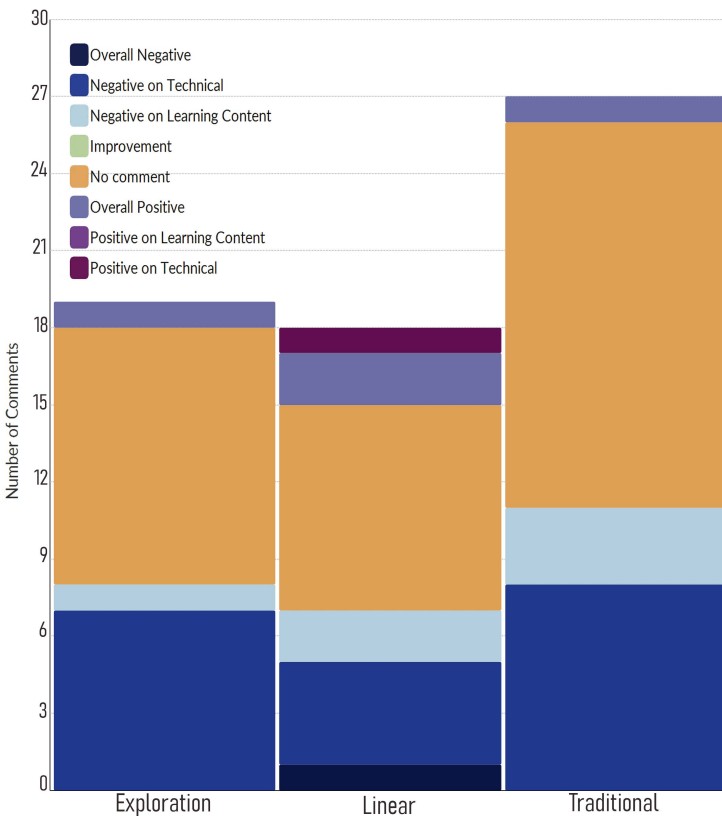

**Figure 8.** Learners' attitudes toward the language-learning session.

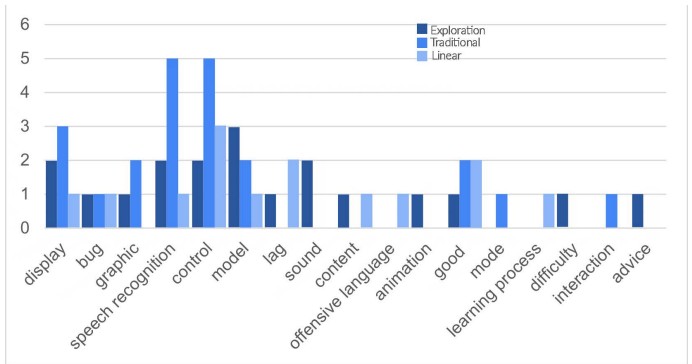

**Figure 9.** The keyword frequency in participants' comments.

## 5. Discussion

This research investigates how different gamification modes in a second-language virtual learning environment affect learners' engagement, learning experience, and perceived learning. Five factors were used to measure the engagement: student-perceived engagement level, student-performed interactions, student-performed positive interactions, student-performed negative interactions, and time spent on learning. This section discusses the quantitative and qualitative results to explore the potential impact of gamified scaffolding on L2 learning and compares the results with past related studies. The acceptance of all null hypotheses in this study reflects an absence of statistically significant differences across experimental conditions. This outcome prompts a reflection on methodological and measurement design. First, there was a limited set of gamification elements used in the experiment. As our study includes avatars, progress bars, and instant feedback, as identified by [54], this set of gamification elements may lack motivational diversity. According to [54], gamification is most effective when multiple elements such as badges, leaderboards, challenges, and dashboards are combined, each targeting distinct motivational mechanisms.

This aligns with findings from [55], who observed only moderate gains in self-regulated learning when similar gamified features were used in isolation. Secondly, our measurement instruments prioritised short-term academic gains. However, as highlighted in [55], some of the most substantial benefits of gamification lie in supporting self-regulation, motivation, and learner autonomy traits that develop incrementally and may not be captured by immediate post-activity tests. Moreover, the homogeneity in results aligns with the study in [26], showing that students with different motivational profiles (e.g., achievers vs. explorers) were not equally engaged, as we did not tailor game elements to player types. This underlines a need for more personalised gamification designs.

### 5.1. Interactivity

This study reveals similar findings compared to previous studies on gamification in language-learning research. Our result confirms the finding from [31] that a virtual learning environment is an effective learning method, yet no significant differences were found between the three different interactivity levels. Ref. [31]'s study also mentioned that higher interactivity groups gained more joyful learning experiences compared to the other two groups. Our results are consistent with this finding, with the perceived learning experience in the higher interactivity level group being slightly higher than others. Learners in the higher-interactivity-level group tended to revisit practices and allocate more time to their learning sessions.

A systematic review of the uses of gamification to support learning revealed that learners have generally expressed a positive experience of gamified learning systems in past related studies [39]. The review also mentioned that most learners reported gamified learning as enjoyable, attractive, and interesting. Their result partially matches the result of our study. In this study, our questionnaire results among the three groups confirm that learners in a more guided gamified environment perceived a greater learning experience.

### 5.2. Motivation and Engagement

A past study by [56] compared the impact of gamified and general learning systems on motivation, engagement, and attention during language learning. Their study revealed that a gamified system significantly increases language learning in three aspects. This result is different from ours, as our study shows that a more gamified system does not significantly increase engagement compared to a less gamified system. However, the differences in the results might be due to the gamification elements used in their system differing from the elements chosen in our prototype. In our study, the game elements used were interactive items, instant feedback assessment, and game-like visual and control design. In contrast, their study focused on using challenges to drive curiosity, motivating students to learn. The differences in the applied gamified elements could lead to different learning results. The difference between the study results might be because game elements such as increasing the flexibility of the game environment, progress control, and more interactive items are not as efficient as challenges in increasing learning engagement and motivation. However, our study strategically selected those elements that aligned most closely with our goal of creating an immersive, interactive learning environment. Thus, our study emphasised the integration of interactive items, instant feedback, and game-like visuals. Commonly used game elements, such as scores and levels, were intentionally excluded to prevent a competitive atmosphere. Yet, the potential of other game elements to engage students is undeniable, and we may consider their inclusion in future studies. Ref. [32] highlighted that when evaluating a gamified system, it is crucial to understand if the system genuinely offers an engaging experience or merely presents an empty facade of a game. To enhance the design, comprehensive design studies should be undertaken to make them more game-

centric. However, the author also noted several adverse effects of gamification, such as receiving rewards for non-essential tasks and even delaying work that might distract users from their initial goals.

*5.3. Limitation and Future Research*

Some of the limitations of this study include the following:

1   Limited learning resources: Another limitation is that as an in-house proof-of-concept prototype, our application has limited learning resources and functionalities for the user, and this could affect user engagement if they were expecting a commercial-grade application experience.

2   Lack of adaptive design: In our study, the gamified environment was not personalised to learners' cognitive styles, motivational drivers, or individual gamification preferences. Future work should include pre-assessments of learner profiles to better match gamification strategies with learner needs.

3   Comprehensive user profile: In addition, the participants recruited were from a very similar demographic background—most of them were university undergraduate students, and a majority of them did not have study experiences in an English-speaking country. This could affect the results because some participants felt the content was less relevant to them. Moreover, not factoring in participants' prior gaming experiences provided a neutral ground to evaluate their engagement and perceptions in the virtual learning environment without biases from past experiences. Future research should incorporate a more diverse participant demographic to mitigate the potential biases and provide robust outcomes.

4   Short-term learning duration: Additionally, this research was designed during a Master's study, which limited the duration of the learning session. While short-term studies like ours offer immediate insights, they might not capture the evolving nature of engagement and perceptions over time. Extended learning sessions in a longitudinal setup may potentially reveal different engagement patterns and perception trends not evident in our short-term investigation. Future research could beneficially extend this framework, applying our findings in a long-term study to explore the longitudinal benefits and provide a comprehensive gamification guide for language-learning environments.

We expect to overcome these problems in our future research by working with industry partners who have commercially available applications and a learning content-related user base.

## 6. Conclusions

This paper presented an investigation on the role of gamification in L2 learning in a virtual learning environment by comparing three learning modes from three aspects: engagement, learning experience, and perceived understanding of learning.

The qualitative and quantitative data collected from the user study were analysed. The results revealed that when learning English in a virtual environment, the modes of learning do not significantly affect users' engagement, learning experiences, or perceived understanding of learning. Nevertheless, there are some interesting indications and insights from the results: there is a positive correlation between the attitude towards learning and the level of interaction. For instance, learners in the traditional group exhibited a negative attitude toward learning as they had the most negative interaction rate, as well as the least engagement and confidence levels. It was also found that learners in the linear group performed the most positive interactions, and exploration learners reported great perceived engagement and learning experience along with high confidence in the learning outcomes.

Based on the findings of our study, we suggest the following guidelines for designing virtual learning environments for language learning:

1. Our study revealed that second-language anxiety is not the key factor affecting the learning experience in any of the three modes in a virtual environment. The result indicated that a gamified virtual learning environment could be a powerful tool for language learning as it has the potential to ease users' second-language anxiety.

2. The results reveal that the traditional linear learning context with fewer interactions had the lowest learning experience in the L2 learning environment among the three groups, while participants in both the linear mode and the exploration mode reported a higher level of learning experiences and engagement. However, since the differences were not statistically significant, fully understanding the role of interaction in L2 virtual learning environments requires further investigation.

3. As our study revealed that different modes in virtual language learning do not significantly affect L2 learning in terms of the learning experience, engagement, and perceived learning, we suggest that such applications can adopt any learning mode or a combination of different modes that best suits the learning content.

4. AI should be integrated strategically to enhance specific pedagogical approaches, such as providing adaptive scaffolding for task-based learning, rather than being viewed as a one-size-fits-all solution. Its value lies in its ability to personalise, adapt, and provide targeted feedback within a well-defined learning strategy, guiding the learning process.

**Author Contributions:** Conceptualization, T.H.; Methodology, R.W.; Writing—original draft, X.J.; Writing—review & editing, C.D.; Supervision, C.R. and T.M. All authors have read and agreed to the published version of the manuscript.

**Funding:** This work is supported by the project titled 'AI and Digital Technology in International Education: Enhancing Teaching Methods and Student Engagement through Online Platforms, Virtual Reality (VR), and Augmented Reality (AR).' The funding code is 102738.

**Institutional Review Board Statement:** The study was conducted in accordance with the Declaration of Helsinki, and approved by the Ethics Committee of Deakin University (protocol code SEBE-2022-10 and 1 March 2022).

**Informed Consent Statement:** Informed consent was obtained from all subjects involved in the study.

**Data Availability Statement:** The original contributions presented in this study are included in the article. Further inquiries can be directed to the corresponding authors.

**Acknowledgments:** The Faculty of SEBE Human Ethics Advisory Group (HEAG) is acknowledged for reviewing this project and confirming its compliance with the National Statement on Ethical Conduct in Human Research 2007 (Updated 2018). The approval period extends for four years, until 3 May 2026.

**Conflicts of Interest:** The authors declare no conflicts of interest.

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
