# Peer review of "AI-Powered Gamified Scaffolding: Transforming Learning in Virtual Learning Environment"

_electronics, doi:10.3390/electronics14132732_

Round 1
Reviewer 1 Report
Comments and Suggestions for Authors
The method is partially explained but lacks detail in crucial areas. The document outlines the use of post-surveys (with 4 general questions, 18 Likert scale questions, and an open-ended question - Table 3) and in-game data collection. However, there’s a significant gap in describing the content of these surveys or the specifics of the in-game data collected. The exclusion of gender is justified with reference to prior research ([27]). The study design involves at least three groups: Linear, Exploration, and Traditional (implied by context). Statistical tests used include K-W ANOVA, One-way ANOVA, and Turkey HSD, along with normality and homogeneity checks. A 95% confidence level was applied, and power calculation was performed (>β=.8). Importantly, all null hypotheses were accepted, suggesting no statistically significant differences between groups overall. Some more explanations about that are needed.
Figure 4 is not clear at all. Please clarify it.
Based on the provided context, there is a very limited comparison to state-of-the-art work. The citation of [27] justifies excluding gender as a variable, indicating awareness of some prior research in VLE for L2 studies. However, there’s no broader discussion of existing literature on gamification, learning engagement measurement, or the effectiveness of different VLE designs (linear vs. exploration). The reference to “gamification theory” is made without citing foundational work in that field.
Details about the development are just provided in a mini-chapter (3.2) without further information. I strongly suggest expanding the development section with a system architecture, the involved devices, the data flow...
Comments on the Quality of English Language
The text reads like a draft; minor polishing would improve clarity. There are also a couple of grammatical errors (e.g., “Accpeted”). Overall, it’s understandable but not polished for publication without editing.
Author Response
Dear reviewer,
Please see the attached response letter, thank you!

Reviewer 2 Report
Comments and Suggestions for Authors
[2. Related work]
1. The section title should be Related Work not "Related works"
2. The references in the section on Gamification in Language Learning are old (I suggest adding some newer ones) but the references in the section on Artificial Intelligence in Education are ridiculously old (2004, 2012, 2015) - considering that the wide use of AI in education began with ChatGPT in 2022. Moreover, the latter section is inadequately short given the topic of the paper. It should be updated and extended.
3. There is a section on Gamification in Language Learning and Artificial Intelligence in Education but there is no section on combining Gamification and AI for Education.
While much less developed theme than the others, there is a number of relevant publication to investigate the various uses of AI+gamification, e.g. for personalized learning: https://ieeexplore.ieee.org/abstract/document/10964191, in AR environments: https://ieeexplore.ieee.org/abstract/document/10902870, to generate gamified exercises: https://www.mdpi.com/2076-3417/14/18/8344.
[3. User Study]
4. "The participants were recruited from social media groups and research recruiting platforms." - how were they recruited? Were they individually approached or reacted to a general invitation (in the latter case, you have to mention participation bias among the study limitations).
5. When and where the study took place?
[4. Results]
6. "After filtering, 100 valid responses remained from 125 received responses" - why some responses were disqualified as invalid?
What were the criteria?
[Limitation and Future Research]
7. I'd prefer the Future Research plans to be stated in a more explicit way.
[Technical remarks]
8. In my opinion, the way prior work is referenced should be corrected:
"[13] defined Task-Based Learning (TBL)..." -> rather: Harden et al. [13] defined Task-Based Learning (TBL)
"[10] considered..." -> as above.
"[28] highlighted" -> as above.
Author Response

(The authors gave the same response as above.)

Round 2
Reviewer 1 Report
Comments and Suggestions for Authors
The Authors addressed all the comments according to the expectations. No further modifications required.